Continental variation in wing pigmentation in Calopteryx damselflies is related to the presence of heterospecifics

Hassall Christopher c.hassall@leeds.ac.uk
School of Biology, University of Leeds , Leeds , UK
Andrew Nigel
Electronic publication date: 2014 Jun 10
Publication date: 2014
Volume: 2
Electronic Location ID: e438
Received 2014 Mar 28; Accepted 2014 May 29
Copyright: © 2014 Hassall
Copyright year: 2014
Copyright holder: Hassall
License: This is an open access article distributed under the terms of the Creative Commons Attribution License, which permits unrestricted use, distribution, reproduction and adaptation in any medium and for any purpose provided that it is properly attributed. For attribution, the original author(s), title, publication source (PeerJ) and either DOI or URL of the article must be cited.
License URL: https://creativecommons.org/licenses/by/4.0/

Keywords: Dragonfly, Odonata, Melanin, North America, Species recognition, Allopatry, Wing, Sympatry, Thermoregulation, Immune function

Funding: British Ecological Society Small Ecological Project Grant ref 2770/3465 Government of Canada Postdoctoral Fellowship Ontario Ministry of Research and Innovation Fellowship The study was funded by a British Ecological Society Small Ecological Project Grant (2770/3465) and CH was supported by a Government of Canada Postdoctoral Fellowship and an Ontario Ministry of Research and Innovation Fellowship. The funders had no role in study design, data collection and analysis, decision to publish, or preparation of the manuscript.

==============================
Wing pigmentation in Calopteryx damselflies, caused by the deposition of melanin, is energetically expensive to produce and enhances predation risk. However, patterns of melanisation are used in species identification, greater pigmentation is an accurate signal of male immune function in at least some species, and there may be a role for pigment in thermoregulation. This study tested two potential hypotheses to explain the presence of, and variation in, this pigmentation based on these three potential benefits using 907 male specimens of Calopteryx maculata collected from 49 sites (34 discrete populations) across the geographical range of the species in North America: (i) pigmentation varies with the presence of the closely related species, Calopteryx aequabilis, and (ii) pigment increases at higher latitudes as would be expected if it enhances thermoregulatory capacity. No gradual latitudinal pattern was observed, as might be expected if pigmentation was involved in thermoregulation. However, strong variation was observed between populations that were sympatric or allopatric with C. aequabilis. This variation was characterised by dark wings through allopatry in the south of the range and then a step change to much lighter wings at the southern border of sympatry. Pigmentation then increased further north into the sympatric zone, finally returning to allopatry levels at the northern range margin. These patterns are qualitatively similar to variation in pigmentation in C. aequabilis, meaning that the data are consistent with what would be expected from convergent character displacement. Overall, the results corroborate recent research that has suggested sexual selection as a primary driver behind the evolution of wing pigmentation in this group.

Introduction

The pigment melanin plays a key role in both colouration of the insect integument and in defence against pathogens. In vertebrates, there is evidence for a cost of carotenoid—but not melanin—based pigmentation (Badyaev & Young, 2004; McGraw & Hill, 2000). However, condition dependence of invertebrate melanin-based colouration indicates an energetic cost in this group (Hooper, Tsubaki & Siva-Jothy, 1999; Talloen, Van Dyck & Lens, 2004), suggesting that there may be a strong effect of taxon on the costs of melanin production (Stoehr, 2006). In addition to biochemical synthesis the presence of additional melanin in the cuticle may result in a cost due to increased predation risk due to higher conspicuousness (Svensson & Friberg, 2007).

Pigmentation has also been linked with thermoregulation. For example, Colias butterflies living at higher altitudes possess darker wings which enable greater absorption of solar energy (Watt, 1968) and thermoregulation has been proposed as a selective pressure which drove the early evolution of insect wings (Kingsolver & Koehl, 1985). The dragonfly Aeshna caerulea (Ström) holds its wings close to the ground in such a way as to create a pocket of warm air close to its body, as well as undergoing temperature-mediated physiological colour change (Sternberg, 1996; Sternberg, 1997). There has been some suggestion that the body colouration of coenagrionid damselflies reflects the limits of their thermal tolerance (Hilfert-Rüppell, 1998). Some equivocal evidence exists for a negative relationship between temperature and the degree of wing pigmentation at a population level in Calopteryx damselflies (Outomuro & Ocharan, 2011), while broader comparative analyses of the Calopterygidae show a strong association of pigmentation with species at higher latitudes (Svensson & Waller, 2013). An interspecific comparison of C. virgo (darkly-pigmented) and C. splendens (less-pigmented) showed that C. virgo emerged earlier in the year and maintained a higher body temperature at lower ambient temperatures (Svensson & Waller, 2013). Thus individuals inhabiting cooler regions or emerging during cooler parts of the flight period may benefit from possessing melanin to assist with the absorption of solar radiation to aid thermoregulation, resulting in a positive correlation between pigmentation and latitude (or a negative correlation with temperature). However, field studies suggest that pigmented wings are frequently cooler than the body temperature, suggesting that no heat transfer is occurring (Tsubaki, Samejima & Siva-Jothy, 2010).

Pigmentation is also thought to play a role in species discrimination, influencing both antagonistic conflict between males, and mate choice in both sexes. Two species-pairs of the damselfly genus Calopteryx, one pair in Europe and one pair in North America, have received particular attention with respect to interspecific interactions. C. virgo and C. splendens overlap greatly in their distributions in Europe, with C. virgo extending slightly further south into the Iberian peninsula and north into Scandinavia, while C. splendens is more common in eastern Europe (Dijkstra & Lewington, 2006). Reciprocal hybridisation has been documented in this pair (Tynkkynen et al., 2008) and species isolation is based on both male and female mate discrimination on wing pigmentation (Svensson et al., 2007). Interactions between males of the two species tend to be dominated by C. virgo, and there is evidence that these interspecific interactions may result from poor species recognition in C. virgo which mistake large-spotted C. splendens for conspecifics (Tynkkynen, Rantala & Suhonen, 2004). A decline in the size of the C. splendens wing spot was found in association with greater C. virgo density (Tynkkynen, Rantala & Suhonen, 2004) and where the two species were sympatric (Honkavaara et al., 2011). Thus this species pair may exhibit agonistic character displacement (Grether et al., 2009). This selection for smaller wing spot size in territorial encounters may be traded-off against a selection for larger wing spot size by female mate choice in C. splendens (Siva-Jothy, 1999).

C. maculata (Beauvois) and C. aequabilis (Say) overlap to a lesser extent in North America. C. maculata is found in the south east of the continent while C. aequabilis occupies a band stretching east–west across the northern part of the continent. There is currently no evidence of hybridisation between C. maculata and C. aequabilis (Mullen & Andrés, 2007), despite the readiness of C. maculata males to form tandems with C. aequabilis females (Waage, 1975). This readiness decreases as C. aequabilis female wing pigmentation decreases, rendering it less similar to the C. maculata female (Waage, 1975). C. aequabilis exhibits considerable variation in the size of the pigmented area of the hind wing in males while female hind and fore wings become lighter in populations that are sympatric with C. maculata (Waage, 1979). Thus this species pair may exhibit reproductive character displacement (Waage, 1979).

Previous studies on patterns of pigmentation in Calopteryx damselflies have focused on its role in specific processes such as antagonistic interactions, sexual signalling and immunology. Studies of geographical variation in pigmentation have been conducted, but these were restricted to the small proportion of the range in the southern UK for Calopteryx splendens/virgo (10 populations, Honkavaara et al., 2011) and the northeast corner of the range of C. maculata/aequabilis (26 populations, Waage, 1979). The broadest study of Calopteryx maculata/aequabilis was still focused on the zone of sympatry between the two species, with few populations sampled far beyond that zone (15 populations, Mullen & Andrés, 2007). This study provides a comprehensive description of variation in pigmentation which permits the comparison of competing hypotheses at a continental scale using 907 C. maculata males from 34 sites across the entire range. With these data, two potential hypotheses for geographical variation in the wing pigmentation of C. maculata males are tested: (i) pigmentation is positively related to latitude (or negatively related to temperature) as predicted by a thermoregulatory mechanism, and (ii) pigmentation varies with range overlap with C. aequabilis as would be expected from character displacement. Under the first hypothesis, it would be expected that wing pigmentation would increase with latitude such that individuals experiencing colder average temperatures exhibited more heat-absorbing pigment in their wings. Under the second hypothesis, we would expect a discontinuity in the latitudinal pattern at the edge of the zone of sympatry. Given the generally dark colour of C. maculata wings, this shift might take the form of a lightening of C. maculata wings to provide greater difference compared to the black wing tip of C. aequabilis.

Methods

A total of 907 male specimens of C. maculata were collected from 49 sites across the entire range of C. maculata. Of these 49 sites, a number of samples were consolidated where sites were <4 km apart (the maximum dispersal distance recorded for a Calopteryx species, by the congener C. virgo Stettmer, 1996) and, hence, not independent, to give 34 samples (Fig. 1, for details, see Table S1). Collections took place between 13 May and 7 August 2010 and mean sample size from each site was 26.7 ± 2.9 (SE). Information on the body size of these specimens can be seen in a previous paper (Hassall, 2013). Specimens were taken from stream sites where almost all individuals are reproductively mature adults (note, however, that very occasional younger individuals occupying stream sites may have less-pigmented wings which could add noise to the data, Kirkton & Schultz, 2001). Wings were dissected from the body as close to the thorax as possible and the right hind leg was removed. The four wings were mounted on transparent, adhesive tape (Scotch Matte Finish Magic Tape). Wings were scanned using the slide scanner on an Epson V500 PHOTO flatbed scanner with fixed exposure at 1200 dpi. The slide scanner contains a light source on the opposite side of the object to the scanner and, hence, measures transparency rather than reflectance.

Figure 1 Map of sites with transparency of wings.

Geographical distribution of Calopteryx maculata (light grey) and Calopteryx aequabilis (hashed region). Points mark the location of sampling sites for C. maculata and the size of the point is proportional to the grayscale value of the wing pigmentation intensity (larger symbol = lighter wings) for areas of sympatry (filled symbols) and allopatry (open symbols). See Fig. 3 for illustrations of wing pigmentation variation.

Due to differences between individuals in the area of wing obscured by the thorax and the accuracy of dissections, all wing images were modified to omit the arculus and all regions before the first cross-veins. Wing pigmentation was measured as the average grayscale value of the wing. Grayscale varies between 0 (black) and 255 (white), hence greater values correspond to lighter, more transparent wings. This value was calculated for each pixel on the image, with an average of 503,647 pixels (±1202 SE) on the fore wing and 496,122 pixels (±1260) on the hind wing. All measurements were carried out in ImageJ (Rasband, 1997–2007). During measurements, any damage to wings was noted and those measurements which could not be accurately quantified were excluded. This resulted in the exclusion of 140 fore wing and 116 hind wing pigmentation measurements (see Table S1 for sample sizes).

Mean monthly temperature was extracted for each of the 34 sampling sites in ArcGIS (v9.2) (ESRI, 2006) using the WORLDCLIM gridded “current conditions” dataset (Haylock et al., 2008). The spatial resolution was 10 arc-minutes and only data for the months in which specimens were collected at each site (May, June, July or August) were used. Sympatry was determined using distributional records from Odonata Central (Abbott, 2007) (Fig. 1). A sample from Ohio which is the most-southern population in the sympatric zone may not be sympatric with C. aequabilis at a local level despite lying within the area bounded by populations of C. aequabilis. However, the population is included as sympatric for the analysis. Ordinal date (days since 1st Jan) was used to measure time of season, with which pigmentation has been shown to vary in Calopteryx dimidiata (Burmeister) (Johnson, 1973).

Response variables, predictors and model residuals were tested for normality using Shapiro–Wilk tests. Transformations were applied where those assumptions were not met. General linear models weighted by the square root of the sample size were constructed in R (R Development Core Team, 2013) with fore wing and hind wing pigmentation as response variables. It is unclear (i) how atmospheric temperature relates to the temperature experienced by aquatic larvae and (ii) what aspect of temperature variation might influence odonate pigmentation. As a result, latitude was used as a general proxy for temperature variation. Furthermore, after transformation for normality, latitude and annual temperature were very highly correlated (r = 0.860, p < 0.001). Latitude, ordinal date on which specimens were collected, and sympatry with C. aequabilis were included as predictors in the GLMs. An interaction between latitude and sympatry was also included to evaluate the effect of range overlap on latitudinal trends. Where specimens were collected across multiple dates, the average of those dates was taken for the sample. Subsets of data corresponding to different regions of the range (i.e., whole range, allopatric range and sympatric range) were analysed using Pearson correlations to investigate whether latitude-pigmentation patterns varied. Similarity between specimens from different geographical locations may be as much a function of their proximity as of any underlying patterns. This should be corrected for using spatial autoregressive models, but only if the residuals of the uncorrected models are spatially autocorrelated (Diniz-Filho, Bini & Hawkins, 2003). The residuals of all models were tested for spatial autocorrelation using SAM v3.0 (Rangel, Diniz-Filho & Bini, 2010).

Supplementary data on the degree of wing pigmentation in C. maculata females and C. aequabilis females are available from a previous study and provide a useful comparison (Waage, 1979). Mean transparency of the dark regions of female C. maculata and C. aequabilis wings, and the proportion of the wing that was pigmented in C. aequabilis males were extracted from Waage (1979), see Figs. 3B–3D. Data extraction was carried out using the GetData Graph Digitizer v2.24 (Fedorov, 2008). Waage used a densitometer to quantify relative transparency between samples and so his absolute measurements are not directly comparable to the present study. However, the data still permit a qualitative comparison of variation between the species. All data for the C. maculata males used in this study are available as electronic supplementary information (Table S1 for site summaries, Table S2 for individual measurements).

Results

Note that throughout the results, wing transparency is used as the measure of colouration and this is inversely related to the degree of pigmentation in the wing. A guide is shown in Fig. 2 to give examples of varying degrees of wing transparency. When individual data were averaged across sites there was a highly significant correlation between hind and fore wing pigmentation (n = 34, r = 0.985, p < 0.001) and so only fore wing data are shown in Figs. 1, 2 and 3A. The results from GLMs demonstrate that sympatry with C. aequabilis and latitude influenced variation in fore and hind wing pigmentation but that the effect of latitude was dependent upon sympatry (Table 1). Residuals of these models were not spatially autocorrelated as revealed by SAM and so no control for spatial autocorrelation was necessary. In the allopatric zone, male C. maculata showed consistently dark wings (Figs. 1 and 2). The exception was a single population in Texas in which individuals had lighter wings. Where the range of C. aequabilis overlaps the range of C. maculata, wings suddenly become lighter and then darken further north in the sympatric zone. There is a significant correlation between latitude and mean population pigmentation for populations of C. maculata when sympatric with C. aequabilis (n = 21, r = −0.743, p < 0.001—note that this negative correlation means increasing pigmentation with latitude) but not for allopatric populations (n = 13, r = −0.280, p = 0.332) or populations as a whole (n = 34, r = 0.150, p = 0.397). The decline in pigmentation does not appear to result from a continuation of any trends from the allopatric region but occurs suddenly.

Figure 2 Variation in wing pigmentation in Calopteryx maculata.

Latitudinal patterns of pigmentation in C. maculata males where populations are allopatric (open circles) and sympatric (closed circles) with C. aequabilis. Error bars are 1SE. The y-axis is arranged to show increasing levels of pigment from bottom to top, but note that the measurement was grayscale where higher values correspond to lighter colours. Displayed on the right are examples of wings with pigmentation intensities corresponding to their locations on the y-axis. See Table S1 for sample sizes and means, and Table S2 for raw data.

Figure 3 Comparison of wing pigmentation variation between Calopteryx maculata and C. aequabilis.

Comparison of variation wing pigmentation between (A) C. maculata males (from the present study), (B) C. aequabilis females (from Waage, 1979), (C) C. maculata males (from Waage, 1979), and (D) C. maculata females (from Waage, 1979). All y-axes represent qualitatively the same trait: pigmentation of the wing, with low values (top of y-axes) corresponding to darker wings and higher values (bottom of y-axes) to greater transparency. However, the measures vary in the way in which they were collected: for C. aequabilis males it is the proportion of the wing length that is unpigmented, for C. aequabilis and C. maculata females it is the transparency of the dark area of the wing (using a densitometer), and for C. maculata males it is the mean grayscale value of a wing scan (see methods above). All error bars are standard errors. Note in all cases that pigmentation is highest at the southern margin of the zone of sympatry with subsequent declines further north.

Table 1 General linear models describing variation in wing pigmentation in Calopteryx maculata.

Parameter estimates from general linear models (weighted by square root of sample size) describing variation in wing pigmentation in Calopteryx maculata males. DF = 1 for all parameters.

	Fore wing pigmentation	Hind wing pigmentation	
	Parameter	t	p	Parameter	t	p	
Ordinal date	−0.0017	−0.298	0.768	−0.0015	−0.367	0.717	
Latitude	−0.3068	−0.994	0.329	−0.0944	−0.428	0.672	
Sympatry	2.8438	5.097	<0.001	2.2590	5.591	<0.001	
Latitude*sympatry	−0.5589	−1.574	0.126	−0.5280	−2.067	0.048	
Radj2			0.547			0.581	

The relative wing pigmentation between geographical locations that has previously been reported for C. maculata females, C. aequabilis females, and C. aequabilis males (Waage, 1979) appears to follow the same geographic pattern as that found in the present study for C. maculata males (Fig. 3). In allopatry, wings tend to be more pigmented (lower transparency). At the southern margin of the sympatric zone wings tend to be less pigmented (higher transparency) and pigmentation increases towards the northern margin of the sympatric zone in all four groups. Note that only qualitative comparisons are possible between the groups due to variation in the methods used to obtain the data, but that these comparisons suggest a consistent pattern.

Discussion

I demonstrate that, contrary to previous assumptions, males of C. maculata vary greatly in their pigmentation and this variation coincides with the presence of a congener, C. aequabilis. Qualitatively similar patterns of pigmentation have been observed in females of the same species and in females of the heterospecific C. aequabilis. Across the entire range, no gradual latitudinal patterns are present in wing pigmentation which would be expected if a covariate of latitude (be it temperature or another variable) was influencing pigmentation. However, within the zone of sympatry with C. aequabilis, wings become progressively darker as latitude increases. This finding not only provides another important component of a well-studied evolutionary system (the C. maculata/aequabilis species pair) but also constitutes a thorough test of intraspecific variation in pigmentation with changing latitude at a continental scale.

It has previously been stated that Calopteryx maculata “… has dark wings and shows little geographic variation in the wing pattern” (Mullen & Andrés, 2007) and that a “… sympatric shift in wing pigmentation was exhibited by females of [C. maculata and C. aequabilis] but only by males in C. aequabilis” (Honkavaara et al., 2011). This originated in a misinterpretation of the work of Waage (1979) who focused only female pigmentation in C. maculata and did not measure that in males, stating that “extent of the dark area in C. maculata males was not measured as their wings are 95–100% pigmented” and “… wings of C. maculata are entirely dark and vary only in size… among the populations sampled” (Waage, 1979, p. 106 and 108, respectively) but did not measure the intensity of pigmentation. However, from a broader comparison of sites, the degree of variation is marked (see the comparison of wing pigmentation levels in Fig. 2) and varies depending on sympatry or allopatry with C. aequabilis. No evidence exists for latitudinal patterns in pigmentation apart from a latitudinal decline in pigmentation through the sympatric zone.

From a previous study it has been shown that C. aequabilis reduces its pigmentation progressively in populations that are located deeper inside the range of C. maculata (wing pigmentation is significantly correlated with latitude, see Table 1 in Waage, 1979) and it has been assumed that this was to enhance species recognition where dark-winged C. maculata were present. However, it appears that the northward increase in pigmentation within the sympatric zone is also present in C. maculata, with qualitatively similar trends in increasing pigmentation further north in both sexes of both C. maculata and C. aequabilis. This pattern is consistent with the existence of convergent character displacement (Grant, 1972) with male wing pigmentation changing to match variations in female C. aequabilis pigmentation intensity and male C. aequabilis wing spot size (see Fig. 3 for a comparison with Waage’s (1979) data). However, a notable difference between the two trends is that the decline in pigmentation in C. maculata males is found in both wings, while this is only true for the hind wing in C. aequabilis (Waage, 1979). This difference may relate to variations between species in the “cross-displays” performed by males to court females, which result in the more prominent display of hind wings in C. aequabilis (Waage, 1973).

It is important to note that the previous demonstration of character displacement involved behavioural differences between species. The key observation was that C. maculata males exhibit a greater ability to discriminate between conspecific and heterospecific females when in sympatry than when in allopatry (Waage, 1975). This can be taken as evidence of a cost of confused mating by C. maculata which leads to a selective pressure acting on the reinforcement of species identification. The present study provides a description of apparent convergence in a character that is key to discriminating between species, which appears to render both sexes of both species less discriminable. This observation runs counter to what would be expected given this selection pressure. Furthermore, the pattern cannot be explained by correlations with latitude or temperature, which might be predicted based on the fact that melanin can play a role in thermoregulation at higher latitudes (leading to a positive correlation between latitude and pigmentation), as the levels of pigmentation return to allopatric levels (dark pigmentation) at the northern-most sites (Figs. 1–3).

The pattern of results effectively appear to rule out temperature and latitude as causal factors in the broader trend, although these have been suggested to influence geographic variation in odonate colour (Hilfert-Rüppell, 1998) and are strongly associated with the occurrence of wing pigmentation across the Calopterygidae as a group (Svensson & Waller, 2013). It is worth noting that this is a purely correlative study, albeit involving intensive sampling of populations across an extensive geographical area. The relatively constant levels of wing transparency through the allopatric zone offer no evidence of a latitudinal or temperature-driven cline in pigmentation, although the results are consistent with such a cline (showing declining transparency further north) within the zone of sympatry. Stream temperatures measured in situ would provide a far closer approximation of the thermal environment within which the animals develop, and there is the potential that Calopteryx sp. may specifically select water bodies with particular thermal regimes (as they do for a large number of other parameters, Siva-Jothy, Gibbons & Pain, 1995). Hence, while latitude and temperature are strongly correlated, small (e.g., catchment) scale variations in water temperature may still be operating. Calopteryx can be maintained in laboratory conditions and so a common garden experiment that examined the impacts of varying temperature on larvae from multiple populations would provide a thorough and robust test of these competing hypotheses. Another mechanism that has been implicated in driving changes in odonate colour is the avoidance of harassment by conspecifics (Van Gossum, Stoks & De Bruyn, 2001) or heterospecifics (Tynkkynen, Rantala & Suhonen, 2004). Evidence for this playing a role in calopterygid damselflies is present in the diversity of wing morphs exhibited by European species. Males of Calopteryx exul (Selys), inhabiting north Africa, have no pigmentation on their wings and so resemble females. Calopteryx virgo virgo has no pigmentation at the base or the tip of the wing and resembles Calopteryx splendens found in eastern Europe. Calopteryx splendens balcanica females have pigmented wings which resemble the male (androchrome). Wing pigmentation polymorphism is also present in males of the calopterygid damselfly Mnais costalis (Selys), where clear-winged males exhibit a “sneaker” mating strategy while orange-winged males are territorial (Plaistow & Tsubaki, 2000). Therefore, there is evidence from the Calopterygidae for male-mimicking females, female-mimicking males and, possibly, heterospecific mimicry. Establishing which, if any, of these explanations best fit the C. maculata/C. aequabilis system would require further experiments along the same lines as those conducted previously (Tynkkynen, Rantala & Suhonen, 2004; Waage, 1979).

A conclusive demonstration of convergent character displacement requires knowledge of both the selection pressures and the evolutionary processes that are causing the convergence of traits. Clearly a case for convergent displacement cannot be made purely on the basis of the observations described here, although the patterns resemble those that would be expected from such a process. Very few examples of convergent character displacement have been documented, despite being theoretically plausible (Abrams, 1996). Among these examples, Leary (2001) found that the nature of the calls given by male toads (Bufo sp) to prevent prolonged amplexus by conspecific and heterospecific males converged when in sympatry. This convergence may facilitate interspecific communication to reduce wasteful energy expenditure and exposure to predation. Following the invasion of American mink (Mustela vison (Schreber)) to Belarus, the larger invading species decreased in size while the native European mink (M. lutreola (L)) increased in size (Sidorovich, Kruuk & Macdonald, 1999). On the other hand, many examples of divergent character displacement have been documented (Dayan & Simberloff, 2005).

The importance of thermoregulatory behaviour has been noted in a number of Odonata (for a review see Hassall & Thompson, 2008), and a number of adaptations are present. There has been a suggestion that variations in body pigmentation with latitude in Orthetrum cancellatum (L) contribute to enhanced absorption of solar energy (Hilfert-Rüppell, 1998). Similarly, a melanic form of Sympetrum striolatum (Charpentier) found only at the northern range margin where such pigmentation could assist in thermoregulation was previously described as a separate species, Sympetrum nigrescens, before molecular studies demonstrated that the two were synonymous (Pilgrim & Von Dohlen, 2007). The only other study of range-wide variation in wing pigmentation in a damselfly (Calopteryx splendens) also demonstrated no consistent patterns with latitude (Sadeghi, Adriaens & Dumont, 2009), and this has been attributed to different wing morphs being genetically distinct gene pools with intermediates resulting from hybridisation (Sadeghi, Kyndt & Dumont, 2010). This study similarly shows no unequivocal evidence of a latitudinal cline, despite the wide climatic range over which the species occurs. However, the increased pigmentation at higher latitudes within the sympatric zone is consistent with a potential role in thermoregulation within this region.

It has been demonstrated that the extent of pigmentation in male wings is an honest indicator of immune function in a number of calopterygid damselflies (Hetaerina americana, Contreras-Garduño, Canales-Lazcano & Córdoba-Aguilar, 2006; Calopteryx splendens, Rantala et al., 2000; Calopteryx splendens xanthostoma, Siva-Jothy, 2000). As such, variation in pigmentation may be under indirect selective pressures acting on immune function. Parasites increase in diversity and abundance closer to the equator (Poulin & Morand, 2000) and there is some evidence that virulence follows a similar pattern (Møller et al., 2009). This selection may result in greater immune function at lower latitudes (e.g., Ardia, 2007). Thus we may expect to observe greater pigmentation at lower latitudes where immune challenge is at its highest. However, damselflies emerging later in the season exhibit greater immune responses (Yourth, Forbes & Smith, 2002) which are associated with higher temperatures (Robb & Forbes, 2005). The melanotic encapsulation involved in the immune response follows the same biochemical pathways as those involved in melanisation of the cuticle (Marmaras, Charalambidis & Zervas, 1996). Thus, a negative correlation between pigmentation and latitude (or a positive correlation with temperature) is predicted by both parasite-mediated selection and plastic responses to temperature, but there is no evidence of this pattern in the present study.

The results presented here fill a gap in the knowledge of geographical patterns of wing pigmentation in a well-studied two-species system. In so doing, two key results present themselves. First, there is geographical pattern in wing pigmentation that would suggest variation in thermoregulatory potential across the range, but pigmentation does not vary consistently with latitude. Second, there is strong evidence for variation in wing pigmentation in male C. maculata in relation to its co-occurrence with a congener, C. aequabilis. This pattern of variation is qualitatively similar to patterns seen in C. maculata females, C. aequabilis females, and C. aequabilis males. This apparent convergence of a character that is important in species identification may contribute to the divergent character displacement observed in species recognition behaviour in the same system (Waage, 1975).

Supplemental Information

Table S1 Site summary details for collections of Calopteryx maculata

Locations and sample sizes for sites at which Calopteryx maculata males were sampled. Nfore and Nhind give the numbers of wings that were sufficiently intact to allow analysis. See Table S2 for full data.

Click here for additional data file.

Table S2 Full individual data for wing pigmentation measurements in Calopteryx maculata

Individual data for measurements of pigmentation in all four wings of 905 male specimens of Calopteryx maculata. See Table S1 for locations of sites. Wing damage is rated on a four-point scale: 1 = undamaged, 2 = slight damage, 3 = very damaged, 4 = not usable.

Click here for additional data file.

I would like to thank Arne Iserbyt, Mary Burnham, Chris Lewis, Shari Sokay, Darrin O’Brien, Fred Sibley, Giff Beaton, George Harp, George Sims, Harris Luckham, Mike Luckham, John Abbott, Joseph Carson, Jeni Eggers and Eliott Porter, Jeffrey Willers, Michael Blust, Marion Dobbs, Mark Musselman, Pat Heithaus, Rick Abad, Ryan Spafford, Steve Hummel, Sarah Richer, Timothy Sesterhenn, William Lamp and Wade Worthen for giving so graciously of their time to assist with collections. Carley Centen provided valuable assistance in the field. Tom Langen provided assistance with logistics. Jonathan Waage, Jason Pither and several anonymous referees provided extremely helpful comments on an earlier draft of the manuscript.

Additional Information and Declarations

Competing Interests

Author Contributions

The author declares there are no competing interests.

Christopher Hassall conceived and designed the experiments, performed the experiments, analyzed the data, contributed reagents/materials/analysis tools, wrote the paper, prepared figures and/or tables, reviewed drafts of the paper.

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
