# Peer review of "Continental variation in wing pigmentation in Calopteryx damselflies is related to the presence of heterospecifics"

_PeerJ, doi:10.7717/peerj.438_

## Round 0.1 · original submission · Major Revisions

Both reviewers have suggested some serious reconsideration of the statistics and interpretations of the data which I agree with, before this manuscript is ready for acceptance - please review each reviewers comments and appraise accordingly.

·

Basic reporting

This manuscript meets all the expectations regarding “basic reporting”.

Experimental design

Field sampling was geographically extensive, locally intensive, and provided for good sample sizes for the purpose of this observational / correlative study.

Validity of the findings

The most reliable conclusions relate to the clear pattern that pigmentation is strongly correlated with latitude within the zone of sympatry, but not elsewhere. This specific pattern is evident enough (Fig. 3) such that formal statistical tests are overkill. However, I have some questions / concerns regarding the stats and interpretations more generally, and these are outlined in the comments to authors.

Additional comments

This is an interesting study, and one that provides important data and information relevant to the field. I do, however, have some general comments / suggestions aimed at improving the manuscript.

I suggest the following minor revisions to the figures to improve legibility:
I printed Fig. 1 on a black&white printer, and I suggest (i) making the background continent basemap being white (with state / province boundaries in light grey), and C. aequabilis range hashed, and C. maculate light grey. Overlapping areas will still be clear. Then (ii) make the sample points that fall exclusively in the C. maculata range (south) hollow, to correspond to the symbols used in Fig. 3. The legend in the figure would therefore only indicate what symbol size means (greyscale). (iii) make the legend heading correspond to the y-axis label in Fig. 3. It is presently confusing “Pigment (greyscale)”. (iv) include graticules or tick-marks on the map indicating latitude, as this is important information, and again corresponds to subsequent figures.

Figure 4 has inconsistencies between y-axis labels and what is in the figure heading. For example, panel (A) y-axis label is “proportion wing unpigmented”, but the heading says the opposite. Please double check that all such errors are corrected. It would be best to remain consistent, e.g. higher values of “pigmentation” should represent darker wings, not the other way around. Likewise, higher values of “transparency” should represent lighter wings. Better yet, define one term early on in the methods (either pigmentation or transparency) and use the same term throughout, where appropriate.

Fig. 4 heading should note which data came from where (Waage’s study or the present one).

Fig. 3 uses standard error bars, whereas Fig. 4 uses 95% confidence limits. I suggest being consistent, but this is just a personal preference.

Other general comments:

1. The description of the statistical analyses would benefit from clarification. If “generalized linear models” (GLMs) were used (line 118), what link function was used? Table 1 includes an adjusted R-square value, which suggests that a “general linear model” (ordinary least squares multiple regression) was used rather than a “generalized” linear model, because R-square values are not as easily calculated for GLMs that use something other than a standard Gaussian error distribution.

2. I note that the error bars in Fig. 3 appear to increase in size with increasing mean, suggesting that either a log-transformation might have been necessary, or perhaps a Poisson link function may have worked in a GLM.

3. Include a statement describing how regression assumptions were assessed (e.g. residual diagnostics, or in case of GLM, testing for overdispersion).

4. I totally agree that WORLDCLIM climate data may not reliably reflect the local site conditions, especially with respect to the aquatic habitat of the nymphs. However, “latitude” is not as useful a variable, because it is simply a proxy for things related to latitude, e.g. temperature, sunlight hours, etc… So, I would strongly recommend that, at the very least, a supplemental information section include temperature as a predictor in lieu of latitude. This seems especially important because thermoregulation figures prominently in the introduction and discussion. I understand that latitude makes for an easier discussion (e.g. regarding areas of sympatry) and more consistent presentation of results (e.g. the map, Fig. 3 and 4). So, if you can demonstrate that, for your sample sites, latitude and temperature are strongly correlated (e.g. r > 0.8), then I accept latitude as a proxy because it simplifies presentation.

5. Table 1: the heading states “parameter estimates”, but no parameter estimates are reported… only “F” and “P” values. I suggest providing two full multiple regression tables, which include estimates of the intercept and coefficients associated with each predictor. Also, “t” values are more traditional than “F” for the tests of parameter estimates (F is used for overall significance of regression).

6. Although the sampling was apparently conducted when local populations were mostly comprised of mature adult males, the timing of sampling (i.e. how long into the local adult flight season) may be especially important to consider. Ordinal date is used to represent this, but given the likely substantial variation in temperature and thus phenology among sample sites, I’m not convinced that it serves its purpose that well. Indeed, the more I reflect on this, the more convinced I am that ordinal date is useless: picture emergence happening in early May at sites in the south, and late June in the north. Ordinal dates, as used in the regression, don’t really reflect anything meaningful here, do they? Wouldn’t age of individual be the more proximately important variable regarding pigmentation? Wing wear has been used as an estimate of age, so it would be helpful to know if pigmentation is related to wing wear. Or use wing-wear as a predictor in the multiple regression.

7. Although the author has described alternative hypotheses regarding geographical variation in wing pigmentation, I recommend presenting these along with clear predictions, AND (at least in the Discussion) clear statements about the scope of inference provided by the present data / study. That is, what do the data clearly show consistency / inconsistency with, and what would be required to provide more convincing evidence? For example, having good local temperature data rather than latitude? Comparing pigmentation among individuals from sites in range of overlap that do and do not have both species?

8. The results can be simplified. Presently, the first section of the results shows correlation results, but the methods never anticipated the use of correlations. Rather, the multiple regression (assuming it was conducted appropriately; please see notes above) provides all that is needed. Specifically, given that “sympatry” is a dummy variable (presumably), then its own significance and significant interaction with latitude shows its importance. All the correlation results (lines 152-154) are redundant. I am OK with lines 158-161, as this simply reinforces the main regression results presented immediately before (lines 156-158).

Specific comments:

Line 132: not sure what this first sentence is meant to refer to. I think it should have “were gathered from Waage (1979)”?

Line 141-143. Are ANOVA results necessary? These were not anticipated in the methods, nor do they show anything that the figures don’t, i.e. substantial variation in pigmentation among sites. If ANOVA results are deemed important, then include a section in the methods. Please see my note above about checking assumptions and apparent relation between variance (size of error bars) and the mean.

Line 173-175 regarding data availability; put in methods.

Line 459, “Waage” missing, I believe.

Reviewer 2 ·

Basic reporting

The basic reporting is sounds and, to the best of my assessment , conforms to PeerJ requirements.

It is clearly written and contains is very inclusive with respect to discussing the results with respect to a range of competing hypotheses. All figures are relevant.

One minor suggestion – the second sentence of the abstract doesn’t make sense to me. It seems to be presenting two very different points.

Experimental design

The research described is original and of a high technical standard.

The research questions addressed focus on testing how well multiple hypotheses are supported by a large dataset comprising observations of wing colouration along a latitudinal gradient.
The paper deals well with the potential confounding of latitude and sympatry. It relies heavily on developing a narrative showing how patterns in pigmentation vary and to what degree they support hypotheses framed at very coarse spatial scales.

The measure of transparency rather than reflectance to assess pigmentation is sound in my opinion. I’m not sure how well it would work with other taxa but the images show how it works well with this group.

Validity of the findings

Although this is not an experimental paper the data set generated and the analyses presented constitute a robust piece of science.

The manuscript is suitably cautious in its interpretation. It provides compelling evidence that there’s no support for the thermoregulation hypothesis while describing a pattern. I don’t think the results “rule out temperature and latitude as causal factors” (L228) because in the end the work is correlative; it certainly doesn’t support them as explaining it but ruling them out as causal factors requires a new set of experiments.

The analyses are appropriate although I think there is some detail missing for the GLMs (df in table 1 and discussions line 160-161). This should be addressed.

---

## Round 0.2 · accepted · Accept

I believe you have done a good job in assessing the reviewers comments and am happy for this to be developed for production.